

# ZFP36 ring finger protein like 1 significantly suppresses human coronavirus OC43 replication

Tooba Momin[1], Andrew Villasenor[1], Amit Singh[1], Mahmoud Darweesh[2,3], Aditi Singh[1] and Mrigendra Rajput[1]

[1] Department of Biology, University of Dayton, Dayton, OH, United States of America
[2] Department of Medical Biochemistry and Microbiology, Uppsala University, Uppsala, Uppsala, Sweden
[3] Department of Microbiology and Immunology, Faculty of Pharmacy, Al-Azhr University, Assiut, Egypt

## ABSTRACT

CCCH-type zinc figure proteins (ZFP) are small cellular proteins that are structurally maintained by zinc ions. Zinc ions coordinate the protein structure in a tetrahedral geometry by binding to cystine-cystine or cysteines-histidine amino acids. ZFP's unique structure enables it to interact with a wide variety of molecules including RNA; thus, ZFP modulates several cellular processes including the host immune response and virus replication. CCCH-type ZFPs have shown their antiviral efficacy against several DNA and RNA viruses. However, their role in the human coronavirus is little explored. We hypothesized that ZFP36L1 also suppresses the human coronavirus. To test our hypothesis, we used OC43 human coronavirus (HCoV) strain in our study. We overexpressed and knockdown ZFP36L1 in HCT-8 cells using lentivirus transduction. Wild type, ZFP36L1 overexpressed, and ZFP36L1 knockdown cells were each infected with HCoV-OC43, and the virus titer in each cell line was measured over 96 hours post-infection (p.i.). Our results show that HCoV-OC43 replication was significantly reduced with ZFP36L1 overexpression while ZFP36L1 knockdown significantly enhanced virus replication. ZFP36L1 knockdown HCT-8 cells started producing infectious virus at 48 hours p.i. which was an earlier timepoint as compared to wild -type and ZFP36L1 overexpressed cells. Wild-type and ZFP36L1 overexpressed HCT-8 cells started producing infectious virus at 72 hours p.i. Overall, the current study showed that overexpression of ZFP36L1 suppressed human coronavirus (OC43) production.

Corresponding author
Mrigendra Rajput,
mrajput1@udayton.edu

## INTRODUCTION

CCCH-type zinc figure proteins (ZFPs) are small cellular proteins that are structurally maintained by zinc ions. Zinc ions coordinate the protein structure in a tetrahedral geometry (*Abbehausen, 2019*; *Hajikhezri et al., 2020*). There are over 40 different types of ZFPs that have been annotated so far (*Hajikhezri et al., 2020*). ZFP's unique structure enables it to interact with a wide variety of molecules such as DNA, RNA, PAR (poly-ADP-ribose), and cellular proteins and thus modulate several cellular processes including host

immune response and virus replication (*Müller et al., 2007*; *Cassandri et al., 2017*; *Takata et al., 2017*; *Tang, Wang & Gao, 2017*; *Meagher et al., 2019*; *Lal, Ullah & Syed, 2020*). Among various ZFPs, the CCCH-type ZFP family contains zinc ions that coordinate protein structure by binding to cystine-cystine or cysteines-histidine amino acids (*Abbehausen, 2019*; *Hajikhezri et al., 2020*). The CCCH-type ZFP family has also been characterized for its antiviral (*Hajikhezri et al., 2020*; *Tang, Wang & Gao, 2017*; *Zhang et al., 2020*; *Guo et al., 2004*; *Zhao et al., 2019*; *Gao, Guo & Goff, 2002*; *Zhu et al., 2020*; *Musah, 2004*; *Chen, Jeng & Lai, 2017*; *Scozzafava et al., 2003*; *Schito et al., 2006*; *Angiolilli et al., 2021*) and immune modulator activity (*Wang et al., 2015*; *Tu et al., 2019*; *Haneklaus et al., 2017*; *Lv et al., 2021*; *Matsushita et al., 2009*; *Wawro, Kochan & Kasza, 2016*; *Uehata & Akira, 2013*; *Chen et al., 2018*; *Mino et al., 2015*; *Fu & Blackshear, 2017*; *Stumpo, Lai & Blackshear, 2010*; *Shrestha, Pun & Park, 2018*; *Kontoyiannis, 2018*).

CCCH-type ZFPs show their antiviral efficacy against several RNA viruses including Influenza A virus (*Tang, Wang & Gao, 2017*), retrovirus (*Gao, Guo & Goff, 2002*; *Zhu et al., 2011*; *Zhu et al., 2017*) filoviruses (*Müller et al., 2007*), and alphavirus such as Sindbis virus, Semliki Forest virus, Ross River virus, and Venezuelan equine encephalitis virus (*Bick et al., 2003*). However, CCCH-type ZFP's role on the human coronavirus is little explored. The current study is designed to evaluate the effect of ZFP36L1, a CCCH-type ZFP, on human coronavirus (HCoV)-OC43 replication.

## MATERIALS & METHODS

### Cells, virus strains and virus propagation

HCT-8 cells (ATCC, Manassas, VA, USA) were cultured in Roswell Park Memorial Institute (RPMI) 1640 Medium (Gibco BRL, Grand Island, NY, USA) and supplemented with 10% heat-inactivated fetal bovine serum (FBS), (ATCC, Manassas, VA, USA), and antibiotic-antimycotic: penicillin 100 units /ml, streptomycin 0.10 mg/ml and amphotericin B 0.25 µg/ml (Sigma-Aldrich, St. Louis, MO, USA). During virus culture, HCT-8 cells were adapted to 1% FBS. HCT-8 cells cultured with RPMI 1640 medium supplemented with 1% FBS were used to grow and subsequently titrate the OC43 human coronavirus (HCoV) stain (ATCC, Manassas, VA, USA).

### Overexpression and knockdown of ZFP36L1

HCT-8 cells were stably overexpressed for ZFP36L1 (NCBI reference sequence: NM_001244701.1) with a green fluorescent protein (GFP) marker using a lentivirus vector. The ZFP36L1 gene containing both tandem zinc finger domains (TZFD) such as TZFD1 and TZFD2 were cloned in a pLV-eGFP plasmid with the help of Vector Builder Inc, IL. To make the lentivirus, pLV-eGFP plasmids containing our gene of interest were co-transfected with VSV-G and packaging plasmids encoding Gag/Pol and Rev in HEK293T cells. After 48 h, the supernatant containing the lentivirus was collected and purified by centrifugation followed by filtration. Purified lentivirus was concentrated using a sucrose gradient ultracentrifugation and this concentrated, purified lentivirus was used in the study.

Similar to ZFP36L1 overexpression, HCT-8 cells were knockdown for ZFP36L1 using ZFP36L1 specific shRNA (GTAACAAGATGCTCAACTATA). The ZFP36L1 shRNA was stably expressed using a lentivirus by cloning it in a pLV-mCherry plasmid. Lentivirus for ZFP36L1-shRNA was prepared as per the above-mentioned method by co-transfection of pLV-mCherry containing ZFP36L1 shRNA with VSV-G and packaging plasmids in HEK293T cells.

The prepared lentiviruses were used to either overexpress or knock down ZFP36L1. Successful lentivirus transduction was measured through GFP or mCherry expression for ZFP36L1 overexpression (GFP) or ZFP36L1knockdown (mCherry), respectively. Transduced HCT-8 cells were selected with an increased concentration of puromycin (2–3 µg/ml) over 7 days. Selected cells were further characterized for ZFP36L1 overexpression or knockdown using a western blot with ZFP36L1-specific antibodies.

## Western blot analysis for ZFP36L1 expression

To confirm ZFP36L1 overexpression or ZFP36L1 knockdown; wild type, ZFP36L1 overexpressed and ZFP36L1 knockdown HCT-8 cells were individually seeded in T25 flasks. When cells reached 75–80% confluency, cells were lysed using a radioimmunoprecipitation assay buffer (RIPA buffer) (Cell Signaling Technology, Danvers, MA, USA) supplemented with protease-phosphatase inhibitor (Cell Signaling Technology, Danvers, MA, USA). Lysates were then centrifuged at $3,000 \times g$ for 15 min at 4 °C. The supernatant was collected and the protein concentration in each supernatant was measured using the Pierce$^{TM}$ BCA Protein Assay Kit (Thermo Fisher Scientific, Waltham, MA, USA). 40 µg cell lysates were separated through 12% resolving SDS PAGE gel. After separation, proteins were transfected onto a polyvinylidene difluoride (PVDF) membrane (Thermo Fisher Scientific, Waltham, MA, USA). The PVDF membrane was blocked with 5% skimmed milk (Sigma-Aldrich, St. Louis, MO, USA) in Tris-buffered saline (TBS) for 1 h at room temperature followed by incubation with anti- ZFP36L1 antibody (1:1000) (Thermo Fisher Scientific, Waltham, MA, USA) and anti- $\beta$actin antibody (1:4000) (Cell Signaling Technology, Danvers, MA, USA) overnight at 4 °C. After overnight incubation, membranes were washed with tris-buffered saline +0.1% Tween 20 (TBST) and incubated with HRP conjugated secondary antibodies (1:2000) for 1 h at room temperature. After washing, membranes were developed using the Pierce ECL Western Blotting Substrate (Thermo Fisher Scientific, Waltham, MA, USA). Images of the western blot were taken by the Odyssey XF Imaging System (LI-COR Biosciences, Lincoln, NE). Band intensity for ZFP36L1 proteins was normalized with $\beta$-actin using ImageJ software (*Schneider, Rasband & Eliceiri, 2012*). A significant difference in ZFP36L1 expression in ZFP36L1 overexpressed and knockdown cells compared to wild-type cells was estimated using a paired $T$-test.

## Determining ZFP36L1's effect on HCT-8 cells viability

The effect of ZFP36L1 overexpression and its knockdown on cell viability was measured by trypan blue exclusion assay (*Strober, 2015*). Wild type, ZFP36L1 overexpressed and ZFP36L1 knockdown cells were individually seeded in 6 well plates ($1.5 \times 10^6$ cell/well) in triplicate. 96 h post-seeding, cells were washed with sterilized phosphate-buffered saline

(PBS) and detached with 0.25% trypsin-EDTA (ATCC, Manassas, VA, USA). Detached cells were washed with PBS by centrifugation at $500 \times g$ for 5 min at 4 °C, and then cells were stained with 0.4% trypan blue for 3 min and examined for cell. Changes in cell viability following ZFP36L1 overexpression or its knockdown compared to wild-type cells was estimated by paired $t$-test.

## Measuring the effect of ZFP36L1 expression on virus titration

Wild type, ZFP36L1 overexpressed and ZFP36L1 knockdown HCT-8 cells were infected with HCoV-OC43 with 0.1 multiple of infection (MOI) individually. The supernatant from these cells was collected at 24 h, 48 h, 72 h, and 96 h p.i. Collected cell supernatants were then centrifuged at $1,000 \times g$ at 4 °C for 15 min to remove cell debris and stored at $-80$ °C until used. Once samples from all time points were collected, the HCoV-OC43 virus titer was determined as per the aforementioned method (Reed & Muench, 1938). Changes in virus titer in ZFP36L1 overexpressed or ZFP36L1 knockdown cells were compared to wild-type cells and statistically analyzed by a paired $T$-test.

## Measuring the effect of ZFP36L1 expression on HCoV-OC43 replication

To measure the effect of ZFP36L1 overexpression or ZFP36L1 knockdown on HCoV-OC43 replication, we infected ZFP36L1 overexpressed, ZFP36L1 knockdown or wild type HCT-8 cells with HCoV-OC43 (MOI: 0.1). Infected cells were collected at 72 and 96 h p.i. Viral RNA was isolated from infected cells using the QIAamp Viral RNA Mini kit (Qiagen, Valencia, CA, USA). The viral nucleocapsid was quantified using qPCR (Stratagene MX3000P Real-Time Thermocycler; Stratagene Inc., La Jolla, CA, USA) in 25 µl reaction using syber green dye. Primer sequence for nucleocapsid (F: 5′-: GCTGTT TWTGTTAAG TCYAAA GT-3′, R: 5′- ATTCTGATAGAGAGTGCYTAT Y-3′) were used (Al-Khannaq et al., 2016) with qPCR amplification cycle at 95 °C/2 min, 40 cycles of (95 °C/15 and 60 °C/1 min) followed by melting curve cycle at: 95 °C/ 15 s, 60 °C/1 min and 95 °C/15 s. Fold change in HCoV nucleocapsid expression in each cell was estimated by paired -test.

### Statistical analysis

The significant change in HCoV-OC43 titer and virus replication in wild-type, ZFP36L1 overexpressed, or ZFP36L1 knockdown cells was estimated using a paired $T$-test with 95% degree of freedom. Virus titer in wild-type, ZFP36L1 overexpression or ZFP36L1 knockdown cells was repeated at least three times with calculations for average, standard deviation, and standard error.

## RESULTS

### ZFP36L1 was overexpressed or knockdown in HCT-8 cells

A stable ZFP36L1 overexpression with an upstream GFP marker in HCT-8 cells was generated using a lentivirus system. GFP expression in HCT-8 cells was considered positive for ZFP36L1 overexpression (Fig. 1B), which was further confirmed by western blot (Figs. 2 and 3). Similarly, ZFP36L1 was knockdown using ZFP36L1-specific shRNA. The shRNA was located downstream to mCherry and expression of ZFP36L1-specific shRNA was
Wild Type HCT-8 cells     ZFP36L1 overexpressed HCT-8 cells with GFP     ZFP36L1 Knockdown HCT-8 cells with mCherry

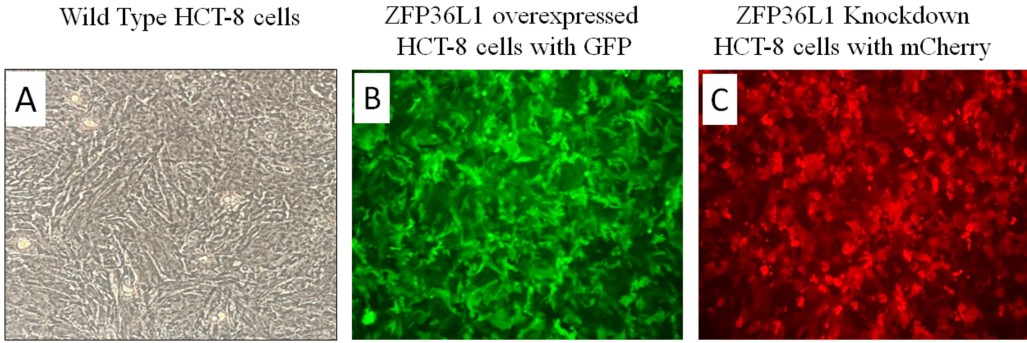

**Figure 1 Overexpression and knockdown of ZFP36L1 in HCT-8 cells.** Wild type HCT8 wells (A), ZFP36L1 overexpressed HCT-8 cells with GFP marker (B), and ZFP36L1 knockdown HCT-8 cells with mCherry marker (C). Overexpression and knockdown of ZFP36L1 were performed by lentivirus transduction.

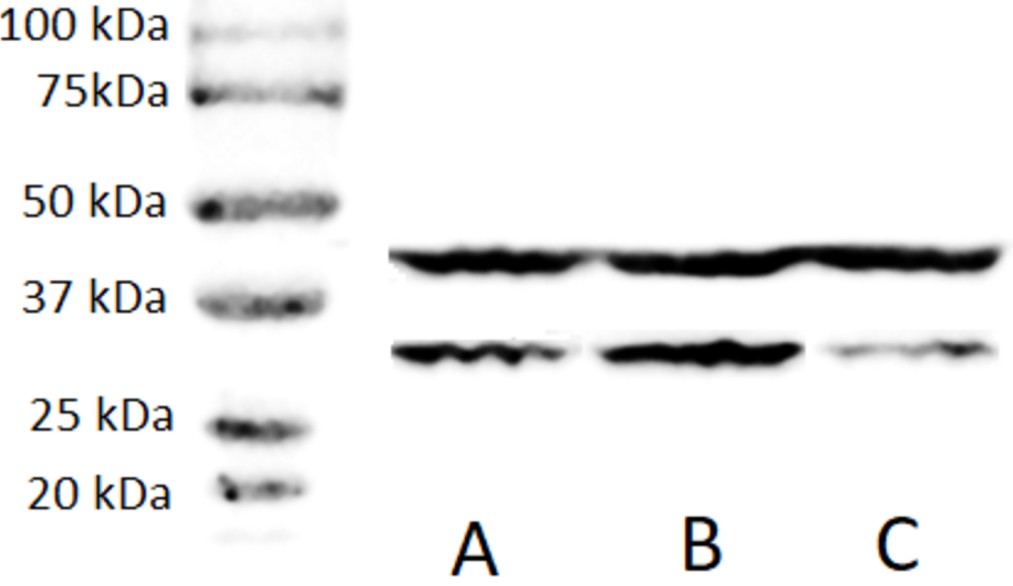

**Figure 2 Western blot for confirming ZFP36L1 overexpression and knockdown in HCT-8 cells.** Cell lysate for wild type HCT-8 cell (A), ZFP36L1 overexpressed (B) and ZFP36L1 knockdown (C) were separated with 12% resolving SDS PAGE gel and transferred to PVDF membrane. Proteins on the membrane were detected with an anti-ZFP36L1 antibody and anti- $\beta$actin antibody with HRP-conjugated secondary antibodies.

determined by mCherry expression (Fig. 1C) and ZFP36L1 knockdown was confirmed by western blot analysis (Figs. 2 and 3). Our results showed that lentivirus significantly overexpressed or knockdown ZFP36L1 in HCT-8 cells ($p < 0.05$) (Figs. 2 and 3).

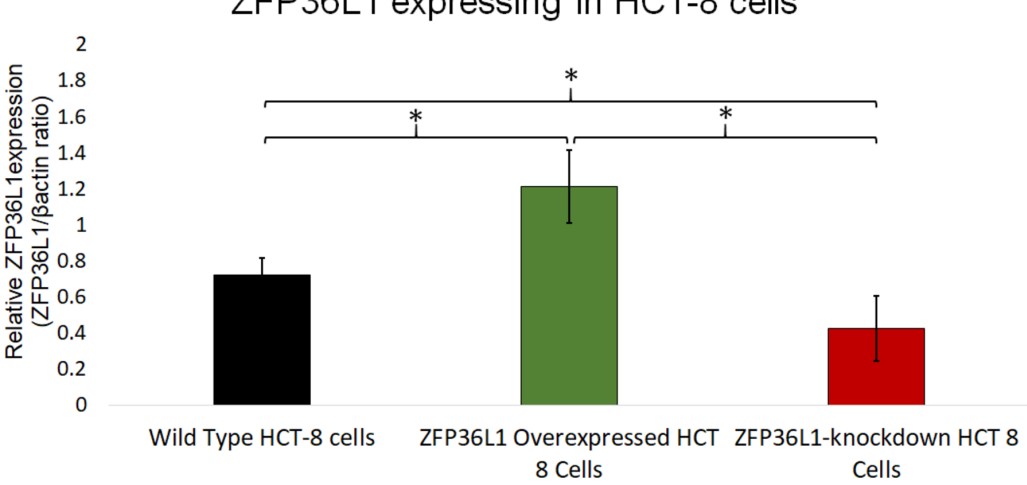

**Figure 3** **Relative quantification of ZFP36L1 expression in HCT-8 cell following its overexpression and knockdown.** Cell lysate for wild-type HCT-8 cell, ZFP36L1 overexpressed and ZFP36L1 knockdown was analyzed for ZFP36L1 and $\beta$ actin using western blot. Band intensity for ZFP36L1 proteins was normalized with $\beta$ actin using ImageJ software. A significant difference in ZFP36L1 expression in ZFP36L1 overexpressed and knockdown cells compared to wild-type cells was estimated using a paired $T$-test. Asterisks show significant differences in ZFP36L1 expression.

## ZFP36L1 overexpressing or its knockdown did not affect HCT-8 cells' viability

The effect of ZFP36L1 overexpression or its knockdown was measured on HCT-8 cells' viability using trypan blue exclusion assay. The results showed that overexpression or knockdown of ZFP36L1 in HCT-8 cells did not affect its viability. Wild type, ZFP36L1 overexpressed and ZFP36L1 knockdown cells showed viability as 94.83 ± 1.01%, 94.16 ± 0.71%, and 95.83 ± 0.43% at 96 h post seeding, respectively These values were non-significant different to each other ($p < 0.05$) (Fig. 4). Additionally, no apparent morphological changes were observed among these cells.

## ZFP36L1 overexpression significantly suppressed, while ZFP36L1 knockdown significantly enhanced the HCoV-OC43 production

Wild type, ZFP36L1 overexpressed, and ZFP36L1 knockdown HCT-8 cells were infected individually with HCoV-OC43 with a MOI of 0.1. Cell supernatants were collected at 24 h, 48 h, 72 h, and 96 h p.i. and analyzed for virus titer.

The results showed that ZFP36L1 overexpression in HCT-8 cells significantly reduced virus titer ($p < 0.05$) (Fig. 5). Virus titer in ZFP36L1 overexpressed cells ion was 2.24 ± 1.28 log 10/ml and 4.32 ± 0.00 log 10/ml at 72 h and 96 h p.i. respectively. These titer values were significantly lower than virus titers in wild-type cells at same time points, such as 72 h p.i. (4.08 ± 0.11 log 10/ml) and 96 h p.i. (5.42 ± 0.10 log 10/ml) ($p < 0.05$) (Fig. 5).

Results with ZFP36L1 knockdown HCT-8 cells showed that ZFP36L1 knockdown significantly enhanced virus titer ($p < 0.05$) (Fig. 5). Knocking down ZFP36L1 facilitated the infectious virus production as early as 48 h p.i. while wild-type cells produced infectious

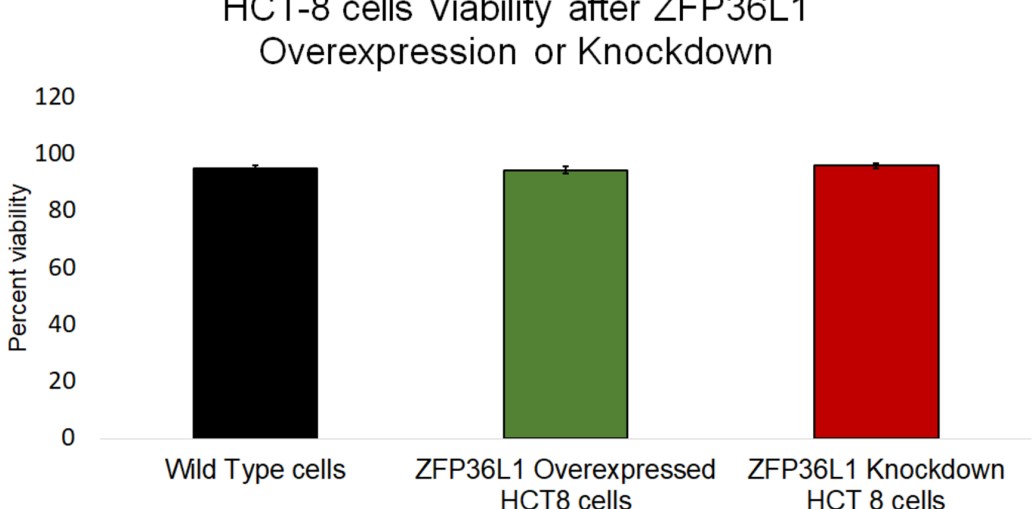

**Figure 4  Effect of ZFP36L1 on HCT-8 cells viability.** The effect of ZFP36L1 overexpression and its knockdown on cell viability was measured by trypan blue exclusion assay. Wild type, ZFP36L1 overexpressed and ZFP36L1 knockdown cells were individually seeded in six well plates. After 96 h post-seeding, cells were detached and stained with 0.4% trypan blue to determine the percent viability. Changes in cell viability following ZFP36L1 overexpression or its knockdown compared to wild-type cells was estimated by paired $T$-test.

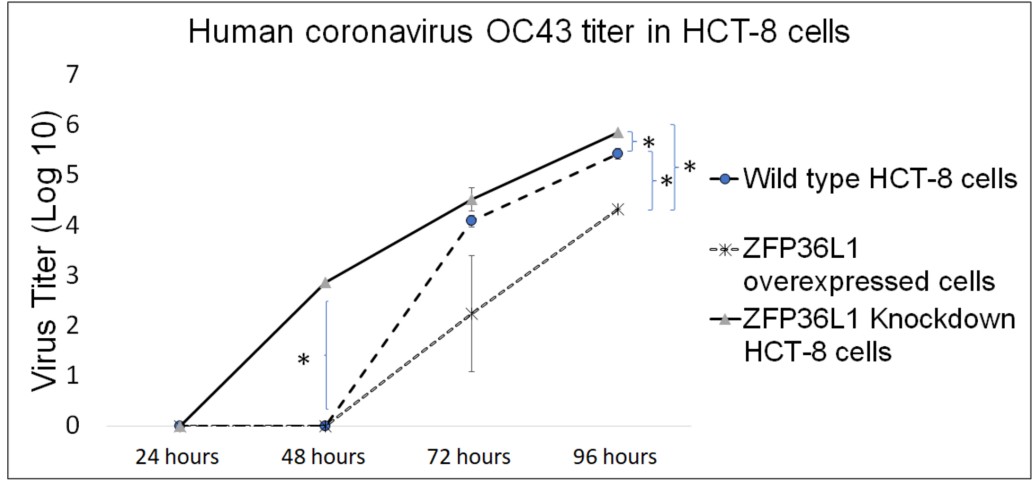

**Figure 5  Human coronavirus-OC43 titer in HCT-8 cells.** Wild type, ZFP36L1 overexpressed and ZFP36L1 knockdown HCT-8 cells were infected individually with HCoV-OC43 with 0.1 MOI. Supernatant from these cells was collected at 24 h, 48 h, 72 h, and 96 h p.i. and analyzed for virus titer. Asterisks show significant differences in virus titer.

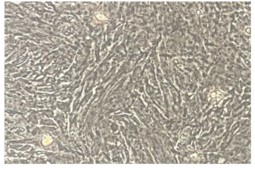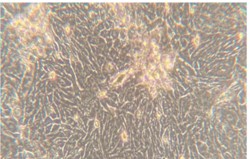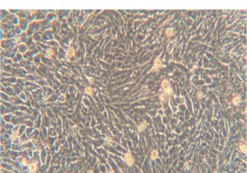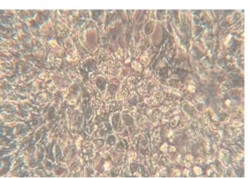

| Mock infected Wild Type HCT-8 cells | HCoV-OC43 infected Wild Type HCT-8 cells | HCoV-OC43 infected ZFP36L1 overexpressed HCT-8 cells | HCoV-OC43 infected ZFP36L1 Knockdown HCT-8 cells |

**Figure 6** **Effect of ZFP36L1 expression on Human coronavirus-OC43 induced cytopathic effect in HCT-8 cells.** Wild type, ZFP36L1 overexpressed and ZFP36L1 knockdown HCT-8 cells were infected individually with HCoV-OC43 with 0.1 MOI. Cytopathic effect was observed at 72 h p.i. at $40 \times$ magnification.

viruses at 72 h p.i. The virus titer in ZFP36L1 knockdown cells was recorded as $00.00 \pm 0.00$ log 10/ml, $2.86 \pm 0.00$ log 10/ml, $4.52 \pm 0.22$ log 10/ml, and $5.85 \pm 0.01$ log 10/ml at 24 h, 48 h, 72 h and 96 h p.i., respectively. While wild-type HCT-8 cells have virus titer of $0.00 \pm 0.00$ log 10/ml, $0.00 \pm 0.00$ log 10/ml, $4.08 \pm 0.11$ log 10/ml, and $5.42 \pm 0.10$ log 10/ml at 24 h, 48 h, 72 h and 96 h p.i., respectively. Virus titer in ZFP36L1 knockdown cells was significantly higher at 48 h and 96 h p.i compared to wild-type cells ($p < 0.05$) (Fig. 5). Results also showed a lower cytopathic effect in ZFP36L1 overexpressed or wild-type HCT-8 cells compared to ZFP36L1 knockdown cells at 72 h p.i. (Fig. 6)

## ZFP36L1 overexpression significantly suppressed while ZFP36L1 knockdown significantly enhanced the HCoV-OC43 RNA replication

To further confirm ZFP36L1's effect on HCoV-OC43 RNA replication, wild type, ZFP36L1 overexpressed and ZFP36L1 knockdown HCT-8 cells were individually infected with HCoV-OC43 (MOI: 0.1). Infected cells were collected at 72 and 96 h p.i. Viral RNA was isolated from infected cells and viral nucleocapsid transcription (RNA concentration) was analyzed using qPCR. Results showed a significant increase in HCoV-OC43 nucleocapsid expression at 72 h and 96 h p.i. in ZFP36L1 knockdown HCT-8 cells while ZFP36L1 overexpression significantly suppressed HCoV-OC43 nucleocapsid expression at both of these time points (*e.g.*, 72 and 96 h p.i.) ($p < 0.05$). ZFP36L1 knockdown HCT-8 cells displayed a $3.56 \pm 0.77$ and $7.67 \pm 1.69$-fold increase in HCoV-OC43 nucleocapsid expression as compared to wild-type HCT-8 cells at 72 h and 96 h p.i., respectively (Fig. 7). ZFP36L1 overexpressed cells displayed a significantly lower HCoV-OC43 nucleocapsid expression such as $0.38 \pm 0.16$ and $0.16 \pm 0.03$ fold compared to wild-type cells at 96 h p.i. ($p < 0.05$) (Fig. 7).

## DISCUSSION

The current study was designed to determine the role of ZFP36L1 (a CCCH type ZFP) on HCoV-OC43 replication. Our results showed that overexpression of ZFP36L1 significantly reduced infectious HCoV-OC43 production while ZFP36L1 knockdown significantly enhanced virus titer compared to wild-type cells. ZFP36L1 overexpression also reduced the

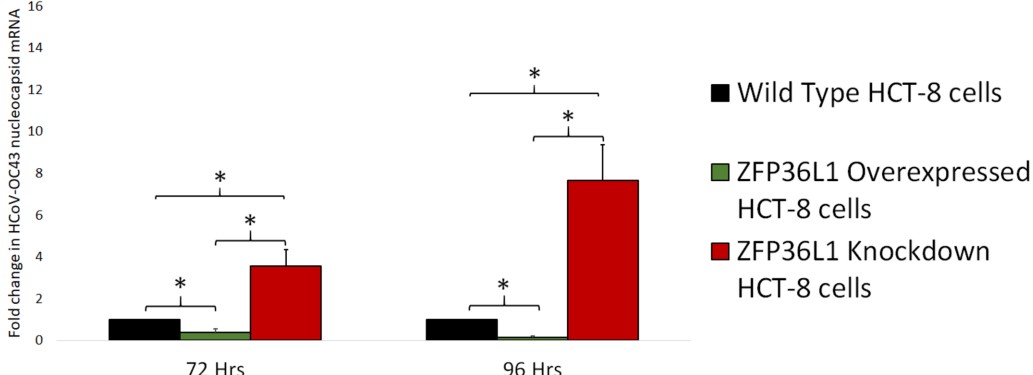

**Figure 7** **Effect of ZFP36L1 expression on Human coronavirus-OC43 replication.** Wild-type, ZFP36L1 overexpressed and ZFP36L1 knockdown HCT-8 cells were infected individually with HCoV-OC43 with 0.1 MOI. Viral RNA was isolated from infected cells at 72 and 96 h p.i. Isolated RNA was quantified using qPCR (for viral nucleocapsid). Fold change in nucleocapsid RNA in ZFP36L1 overexpressed and knock-down cells as compared to wild-type HCT-8 cells were estimated using paired $T$-test. Asterisks show significant differences in viral RNA.

RNA replication of HCoV-OC43 and suppressed the apparent cytopathic effect in infected cells.

ZFPs are one of the most abundant proteins in humans which can make up to 5% of total human proteins (*Vilas et al., 2018*). ZFPs have an extremely high binding ability. They can bind to cellular DNA, RNA, lipids, proteins, and PAR (poly-ADP-ribose); therefore, ZFPs can modulate several cellular types of machinery (*Müller et al., 2007*; *Cassandri et al., 2017*; *Takata et al., 2017*; *Tang, Wang & Gao, 2017*; *Vilas et al., 2018*; *Meagher et al., 2019*; *Lal, Ullah & Syed, 2020*). The diverse binding properties of ZFPs make it difficult to characterize their functional effect in cells (*Vilas et al., 2018*). However, such a challenge is overcome by classifying the ZFPs and then identifying their functional characteristics (*Cassandri et al., 2017*). Classification of ZFP is based on zinc ion, zinc ion interaction with specific amino acids, and the protein's folded structure (*Krishna, Majumdar & Grishin, 2003*). Based on such classification, CCCH-type ZFP is characterized to interact with RNA and thus modulate RNA metabolism in the cell (*Maeda & Akira, 2017*) including interfering with RNA virus replication (*Gao, Guo & Goff, 2002*; *Cassandri et al., 2017*).

The known mechanisms by which CCCH-type ZFPs exhibit these antiviral or immunomodulatory activities is by limiting the total mRNA turnover in the cell. CCCH-type ZFPs such as ZFP36L1 have two tandem zinc finger (TZF) domains that are known to bind with adenyl and uracyl nucleotides-rich (AU-rich) elements (AREs) in mRNA. This interaction facilitates RNA degradation by CCR4-NOT complex-mediated deadenylation, followed by 5′ decapping and exonuclease-mediated nucleotide cleaving (*Blackshear, 2002*; *Lai, Kennington & Blackshear , 2003*; *Lykke-Andersen & Wagner, 2005*; *Suk et al., 2018*; *Lai et al., 2019*; *Lai et al., 2000*).

Coronavirus genome, including HCoV-OC43's genome is 5′-capped with a 3′ poly(A) tail of variable length (*Fehr & Perlman, 2015*). The length of the poly (A) tail varies at different

stages of the virus replication cycle and viruses with longer poly (A) tails replicate at a faster rate (*Wu et al., 2013*). Therefore, the effect of ZFP36L1 on viral poly (A) may explain reduced virus production with ZFP36L1 overexpression in the current study. Our study not only showed that ZFP36L1 suppressed the infectious HCoV-OC43 production, but also reduced HCoV-OC43 nucleocapsid transcription indicating that ZFP36L1 mediates its antiviral effect by limiting the viral RNA in infected cells.

However, there is the possibility that ZFP36L1 can reduce virus replication with different mechanisms other than poly A tail interaction. A study showed that CCCH Type ZFP also targets the non-ARE sequence of 3′ and 5′ (untranslated region) UTR in mRNA (*Li et al., 2015*). Another study showed that CCCH Type ZFP targets CG-rich viral sequences (*Meagher et al., 2019*). The study also showed that ZFP36 (ZFP36L1) suppressed the virus production (influenza A virus) by interfering with viral protein translation/export from the nucleus to the cytoplasm without affecting viral RNA replication (*Lin et al., 2020*). Therefore, a detailed study to determine ZFP36L1's mechanism of action for suppressing coronavirus replication needs to be explored.

## CONCLUSIONS

The current study showed that overexpression of ZFP36L1, a CCCH type ZFP significantly reduced HCoV-OC43 RNA (nucleocapsid) and infectious virus production. A reduced viral production was in correlation with reduced cytopathic effect in the infected cells. Furthermore, ZFP36L1 knockdown significantly enhanced the HCoV-OC43 replication and infectious virus production. However, additional mechanisms employed to reduce virus replication still need to be explored.

### Funding
This work didn't receive any external funding.

### Competing Interests
The authors declare there are no competing interests.

### Author Contributions
- Tooba Momin performed the experiments, prepared figures and/or tables, authored or reviewed drafts of the article, and approved the final draft.
- Andrew Villasenor performed the experiments, prepared figures and/or tables, authored or reviewed drafts of the article, and approved the final draft.
- Amit Singh conceived and designed the experiments, analyzed the data, authored or reviewed drafts of the article, and approved the final draft.
- Mahmoud Darweesh conceived and designed the experiments, analyzed the data, authored or reviewed drafts of the article, and approved the final draft.
- Aditi Singh performed the experiments, analyzed the data, authored or reviewed drafts of the article, and approved the final draft.

- Mrigendra Rajput conceived and designed the experiments, performed the experiments, analyzed the data, prepared figures and/or tables, authored or reviewed drafts of the article, and approved the final draft.

## Data Availability

The raw data is available in the Supplementary File.

## Supplemental Information

Supplemental information for this article can be found online at http://dx.doi.org/10.7717/peerj.14776#supplemental-information.

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
