# Peer review of "ZFP36 ring finger protein like 1 significantly suppresses human coronavirus OC43 replication"

_PeerJ, doi:10.7717/peerj.14776_

## Round 0.1 · original submission · Minor Revisions

Dear Authors,

As per the comments by our expert reviewers, the concept and planning of the study carried out was appropriate but it has few queries or objections to be addressed before further processing.

Please incorporate all the suggestions or justify your stand (in case, if you do not agree with reviewer's opinion/s.).

Please do the needful and do submit at earliest to avoid further delay.

Best of luck.

Reviewer 1 ·

Basic reporting

The manuscript provides the useful insights in the regulation of CCCH-type Zinc figure protein i.e. ZFP36L1 in human coronavirus OC43 replication. However, the manuscript needs to address the following points to improve its quality.
1. As in line number 48 “Wild type, ZFP36L1 overexpressed, or ZFP36L1 knockdown cells” the authors have used the word ‘or’ but I think ‘and’ will be more appropriate to write. As you have discussed your findings in context with all the three mentioned conditions change or with and in all the appropriate places in the manuscript.
2. Mention post-infection (p.i.) only at its first occurrence in rest all of the manuscript use p.i. only.
3. The source of procurement of FBS is not mentioned. Are you sure that your FBS is free of antibodies. If yes, how? Have you tested it using dot-blot or western blot?
4. Relative expression changes in the level of ZFP36L1 proteins can be assessed after normalization with β-actin using suitable softwares like ImageJ. Please do the analysis so that it can strengthen the quality of manuscript.
5. Line no. 124 ‘h’ needs to be replaced by ‘hour’.
6. Your discussion needs to be in a more descriptive and elaborative form. It seems like to be the replica of your introduction part only. Discuss and explain your research findings in reference to current status of the research.
7. Line no. 181 “ZFPS are one of” needs to be replaced by “ZFPs are one of”
8. Reference section: Mention all the references in same format. In some the journal name is in full whereas for others it is in short form. Maintain the format style as per the guidelines.

Experimental design

no comments

Validity of the findings

no comment

Reviewer 2 ·

Basic reporting

The manuscript contains a great number of mistakes, flaws, and incompatibilities with Instructions for Authors and the reference style guide, is still highly verbose and redundant in style, statements are made which are based on mere speculation, the wording of the abstract is confusing,
Even more problematic are the facts that
Line: 75-76: Authors have mentioned the mechanism of action of ZFP's but lack experiments to calculate mRNA turn over.
CoV plaque-forming units (PFUs) were not quantified in virus specimens, including viral stocks prepared from infected cell culture supernatants quantifying concentrations of replication-competent lytic virions
To determine the extent of inhibition, it is essential to compare the viral copy numbers before and after ZFP36L1 knockdown.

Experimental design

In the abstract, the main objective and novelty of the work should be described succinctly, along with the reason or motivation for undertaking the work (working hypothesis), the main methods of investigation (but without experimental details), the main result(s) (again, without specific data), and the conclusions and relevance for the general field of study in light of the international state of knowledge.
The introduction must be condensed to no more than half, avoiding all these well-known facts or facts that have little to do with the experiments presented here. In the introduction, the state of knowledge relevant to the particular field of research addressed in the manuscript should be presented concisely in a scientific (not "journalistic") style.

Validity of the findings

More experiments are required to support the claim that ZFP36 ring finger protein like 1 significantly
suppresses human coronavirus OC43 replication. viz (Viral nucleic acid analysis ( to ascertain viral nucleic acid decay by ZFP's), ZFP36 and Viral RNA co-immunoprecipitation, Quantitative RT-PCR etc.

Reviewer 3 ·

Basic reporting

No comments

Experimental design

The method section should have included a detailed description of the generation of lentivirus carrying siRNA against ZFP36L1.

Validity of the findings

No comments

Additional comments

1. Please check for the typing error
2. Based on the results, it looks like the 'knockdown of ZFP36L1 relatively enhances the early viral replication and does not have an effect on later points. Please consider including such a statement if that seems logical.
3. Should have included results showing the effect of knockdown of ZFP36L1 on cell viability.

Annotated reviews are not available for download in order to protect the identity of reviewers who chose to remain anonymous.

---

## Round 0.2 · Minor Revisions

Dear authors,

Our expert reviewers appreciated your efforts in improving the quality of the manuscript. However, one of the reviewers has suggested a few more points to be addressed (please find them in the attached document).

So please do the needful and resubmit the manuscript asap.

Best of luck.

Reviewer 2 ·

Basic reporting

All suggestions incorporated

Experimental design

All suggestions incorporated

Validity of the findings

All suggestions incorporated

·

Basic reporting

Article is very well presented. The references supporting a particular work is more than sufficient which can be reduced.

Experimental design

Very well designed and is self explanatory. Methodolgy adopted is well explained.

Validity of the findings

No comments

---

## Round 0.3 · accepted · Accept

Dear Dr. Rajput,

It is my pleasure to inform you that as per the recommendation of our expert reviewers, the manuscript "ZFP36 ring finger protein like 1 significantly suppresses human coronavirus OC43 replication" - has been Accepted for publication in PeerJ.

This is an editorial acceptance and you will be intimated for the list of further tasks before publication. So, I request you to be available for a few days to make the necessary things asap.

Regards and good luck with your future submissions.